# The Emergence of Edible and Food-Application Coatings for Food Packaging: A Review

**DOI:** 10.3390/molecules27175604

**Published:** 2022-08-31

**Authors:** Luk Jun Lam Iversen, Kobun Rovina, Joseph Merillyn Vonnie, Patricia Matanjun, Kana Husna Erna, Nasir Md Nur ‘Aqilah, Wen Xia Ling Felicia, Andree Alexander Funk

**Affiliations:** 1Faculty of Food Science and Nutrition, Universiti Malaysia Sabah, Kota Kinabalu 88400, Sabah, Malaysia; 2Rural Development Corporation, Level 2, Wisma Pertanian, Locked Bag 86, Kota Kinabalu 88998, Sabah, Malaysia

**Keywords:** biodegradable packaging, polysaccharides, molecules, surface coating, preservation, food quality

## Abstract

Food packaging was not as important in the past as it is now, because the world has more people but fewer food resources. Food packaging will become more prevalent and go from being a nice-to-have to an essential feature of modern life. Food packaging has grown to be an important industry sector in today’s world of more people and more food. Food packaging innovation faces significant challenges in extending perishable food products’ shelf life and contributing to meeting daily nutrient requirements as people nowadays are searching for foods that offer additional health advantages. Modern food preservation techniques have two objectives: process viability and safe, environmentally friendly end products. Long-term storage techniques can include the use of edible coatings and films. This article gives a succinct overview of the supplies and procedures used to coat food products with conventional packaging films and coatings. The key findings summarizing the biodegradable packaging materials are emphasized for their ability to prolong the freshness and flavor of a wide range of food items; films and edible coatings are highlighted as viable alternatives to traditional packaging methods. We discuss the safety concerns and opportunities presented by applying edible films and coatings, allowing it to be used as quality indicators for time-sensitive foods.

## 1. Introduction

In recent years, edible food packaging has advanced significantly, which is great news for those seeking to improve their standard of living. People are becoming aware of the significance of food packaging for preserving quality. The primary functions of food packaging are to protect its contents from spoilage caused by microorganisms and other organisms, preserve its quality and safety, and extend the product’s shelf life. It also allows for the commercialization and distribution of the product and contains the required product information [1,2]. Historically, goods were packaged with various materials, including paper, cardboard, metal, glass, and plastic. Diverse external factors, such as the expansion of global food markets, government regulations, the accessibility of raw materials, and consumer preferences, have caused the asx food packaging industry to be constantly in flux [3].

However, this traditional preservation method generates most of the municipal solid waste (MSW). The majority of municipal solid waste (MSW) is produced by this traditional preservation method. The Ministry of Environment and Natural Resources (SEMARNAT) estimates that the daily MSW production in Mexico is 102,895 tonnes, with paper, cardboard, glass, and metals such as aluminum constituting most of this waste. By 2025, it is anticipated that 2.2 billion tonnes of municipal solid waste will have been generated, up from 1.3 billion tonnes in 2012. Common knowledge holds that packaging non-renewable and non-biodegradable materials negatively affects the environment. There is widespread agreement among consumers and environmentalists that they contribute significantly to trash and pollution in the environment [4]. Businesses and academics have developed novel packaging strategies to address this issue, using the plethora of biodegradable packaging materials made from renewable crude [5]. According to Akelah et al. [6], the rising interest in edible packaging is in part attributable to the trend toward improving food quality with edible barriers and the rising demand among consumers for highly processed, fresh-tasting, and long-lasting foods.

Food-safe films and coatings are thin, easily removed layers (0.3 mm) that can be eaten whole or used as a topping. Therefore, the formulation must use only ingredients that follow all local, state, and federal laws and regulations about food. The coatings or films applied to food products must not diminish their flavor or texture. Thin coatings on foods or continuous layers between sections or ingredients of different products are examples of edible packaging. Several obstacles can be encountered when attempting to market food that can be remedied with edible films and coatings. These functions include delaying the movement of water, gas, solvents, and oils, improving structural stability, trapping volatile flavor compounds, and transporting food additives. Aesthetically, they can do wonders by boosting shine, minimizing damage, and masking scars [7]. For instance, edible collagen casings for sausages and hot-melt paraffin wax were used to prevent citrus fruits from absorbing moisture. Apples were covered in wax to make them shiny and keep them from being damaged. The shared properties of coatings’ innovation-based natural polymers can be determined by analyzing the product’s unique characteristics and how they change during production, shipping, and depository. Despite a physical barricade, edible coatings on food products must be packaged in non-edible materials due to contamination concerns. By replacing non-edible materials with edible films and coatings, waste and the environmental impact associated with food packaging can be diminished [8].

Edible coating and edible film can be considered synonyms depending on the context. Unlike edible coatings applied to food items, edible films can function independently as packaging materials. In contrast to edible coatings, edible films maintain their shape well enough to be used as independent packaging materials. To that end, using a variety of gelling agents allows for the production of edible films and coatings with a wide range of qualities [9]. Additionally, manufacturers require cutting-edge materials that facilitate innovative approaches to working with food and packaging. Coatings play an essential role in these circumstances by acting as barriers to prevent the ingress of contaminants and the deterioration of the underlying surface due to oxidation, corrosion, and mechanical stress. Nonetheless, finding coatings with good surface adhesion is essential to guarantee long-term performance. This paper examines the most recent data on edible coatings in the food industry to gain a deeper understanding of the topic. The definitions of the most significant terms contribute to a dynamic taxonomy of shifts in food packaging-related thought. Highlights will include discussions of mass transfer, coating strategies, research and innovation initiatives for coated food products, active components incorporated into compositions, and the boundary properties of coatings. Future developments are also considered to identify knowledge gaps and suggest new research directions. This commentary provides a concise overview of the fundamentals of the coating process, enabling researchers to develop novel edible films and coatings that improve the efficiency and effectiveness of food packaging.

## 2. Current Food Packaging Trends

Food safety issues have become much more prevalent in recent decades, raising public awareness and concern. The food industry is implementing measures to meet the rising demand for safe food supplies. There are new technologies for detecting dangerous pathogens and regulations regarding the cultivation, transportation, and packaging of food. Packaging food with materials derived from renewable resources is becoming increasingly common. Compostable materials in packaging and food service are an emerging trend.

### 2.1. Conventional Packaging

Traditional packaging has been and continues to be widely used across numerous goods and industries, especially in the food industry. Packaging has a significant impact on consumers, particularly when purchasing food. When designing packaging, color, shape, material, and size are prioritized over other product characteristics, influencing how consumers perceive a product. The conventional packaging materials are paperboards, polyethylene films, glass jars, and metal cans. In contrast, eco-friendly and sustainable packaging has largely replaced conventional packaging over the past decade. Consumers are becoming more aware of the side effects of conventional packaging. The original intent of food packaging was to protect the contents from outside elements and lengthen the life of perishable goods. Traditional methods for determining whether food is still fresh, such as tasting and smelling it, have become obsolete due to packaging innovations as summarize in Figure 1.

#### 2.1.1. Petroleum-Based Plastics

Metal and glass were traditionally used to package food items centuries ago because they are durable and effective at extending the shelf lives of the food they contain. The advent of petrochemical plastics transformed the packaging of food consumed by the food industry over time. Plastics derived from petroleum, such as polyethylene (PE), polypropylene (PP), and polyethylene terephthalate (PET), are currently some of the most frequently used packaging materials. Petroleum plastics are favored over other packaging materials for their light weight, flexibility, portability, and durability [10].

The food industry utilizes a variety of plastics for various purposes. The most prominent and commonly used plastic is polyethylene. Low-density polyethylene (LDPE) and high-density polyethylene (HDPE) are the two variations of polyethylene. LDPE has greater levels of branching within the PE chains, whereas HDPE has much less branching [11]. The food industry frequently chooses LDPE as a packaging material due to its high resistance to moisture permeability and resistance to mechanical and chemical abrasions [12]. In addition to its low cost and high mechanical strength, the material’s poor barrier against gaseous permeation may limit its application in the food industry. Polypropylene is a widely used plastic in the food industry due to its high resistance to thermal degradation. Microwavable food containers are typically made of polypropylene (PP) to provide consumers with the convenience of heating their food for immediate consumption without needing additional tableware [13]. In addition to its low cost and low moisture permeability, PP is widely used because it has a high tolerance to very high temperatures and aggressive chemicals [14]. However, a very high tolerance for extreme heat may limit its applications. To prevent the accumulation of PP in landfills, its industrial applications will be restricted due to its resistance to thermal degradation [15].

Polyethylene terephthalate is another popular petroleum-based plastic commonly used in beverage containers [16]. Polyethylene terephthalate is also renowned for its sturdiness, structural integrity, gas and moisture barrier properties, resistance to high temperatures, and low cost [17]. Polyethylene terephthalate’s applications are not limited to beverages, and most liquid-based foods are packaged in polyethylene terephthalate containers to prevent leakage or contamination. Polystyrene is an extremely lightweight plastic with a wide range of food packaging applications. The aromatic polymer synthesizes each styrene monomer to form a long polymer [18]. PS has become one of the most sought-after packaging materials in the food industry [19] due to its high thermal resistance, excellent moisture-barrier properties, low density, ease of production, and high affordability. The use of PS began to decline when the public realized that the toxic substances released during degradation could harm humans and the environment [20,21]. The high cost required to recycle and process the waste of PS is also a concern because of the highly stable structures of PS as a polymer.

#### 2.1.2. Paper and Cardboard

In addition to plastics, paper and cardboard are also widely used in food packaging. Offering the same benefits as plastic packaging, paper packaging, and cardboard packaging, it provides an environmentally friendly alternative for a different type of packaging material. However, there are disadvantages to using paper and cardboard for food packaging, such as a lack of mechanical strength and a high gas and moisture permeability.

Carl F. Dahl, a German-born chemist who pioneered the pulping technique using sodium sulfate, is credited with the invention of Kraft paper. Using the sodium sulfate pulping method, it was possible to produce Kraft papers with robust and coarse structures. Frequently, one side of Kraft paper is coated with a glossy layer that provides superior moisture and gas-barrier properties. The opposite side, however, would be left uncoated [22]. Typically, the food will come into contact with the glossy layer to prevent the paper from absorbing liquid from the food and losing its structural integrity. Furthermore, the use of waxed paper in food packaging is not something out of the ordinary. The moisture and gas retention properties of wax paper are well-known. They are so inexpensive that they can be utilized on a large scale. However, since wax is used, the possibility of wax cracking on the paper is a concern, especially when the wax paper is exposed to low temperatures and high mechanical pressures. Wax papers are therefore rarely used in flexible packaging such as paper bags. Instead, they are incorporated into boxes and cartons made of cardboard or paperboard to prevent food products from leaking or becoming contaminated [22]. In addition, the use of food-grade paperboard in food packaging has skyrocketed due to its light weight and excellent mechanical resistance to mild pressures. The grammage of paperboards is significantly greater than the grammage of ordinary paper, which is over 250 gsm [22]. Consequently, paperboards are typically used to create boxes and cartons that can hold a variety of food products, whether they are liquid or solid.

#### 2.1.3. Metal

The application of steel, aluminum, and tin in food packaging began long ago. It has been demonstrated that these metals provide superior protection against foreign contaminants and mechanical damage. If these materials were inapplicable, the concept of canned food would be impossible. However, there are a few unavoidable disadvantages to using metals in food packaging. The primary cause is the occurrence of rust, which is followed by the metals’ tendency to corrode when exposed to food. Not all foods are resistant to acids and alkalis, and certain metals can only hold slightly acidic or alkaline foods. A further disadvantage would be the need to thoroughly process canned foods to prevent packaging deformities caused by microbial contamination and improper headspace management [23].

#### 2.1.4. Glass

Glass-based food packaging is also a prevalent type of food packaging material. It has excellent resistance to mechanical pressures and chemicals, low erosion and degradation rates, and low gas and odor permeability. Its functions are similar to those of metal-based food packaging, and the canning process could be applied to glass-based packaging as well. Despite this, glass is commonly used for photostable food products [24], which do not degrade when exposed to light. Glass jars appear to increase the marketability of the product due to their gloss and shine, which may also enhance the product’s overall aesthetics. However, because glass-based packaging is cumbersome and may cause logistical issues in terms of transportation and distribution, many food companies opt for lighter-weight packaging alternatives. The tempering of glass and production of a uniform and standardized packaging line require significant time, labor, and capital expenditures [24].

### 2.2. Biodegradable Packaging

Biodegradable packaging is any type of packaging that can be broken down and decomposed through natural processes. There are numerous advantages to utilizing biodegradable packaging. First, it is eco-friendly because it does not utilize non-renewable resources, unlike conventional petroleum-based plastics. Biodegradable packaging is also beneficial to the environment because it reduces the amount of waste in landfills, which in turn reduces the greenhouse gas emissions from landfills. Biodegradable packaging reduces waste in oceans, rivers, and other bodies of water because it breaks down into smaller pieces and does not float, unlike some low-density plastics.

In place of non-biodegradable plastics, biopolymers are an alternative biomaterial for compostable or biodegradable packaging that can be used to reduce environmental impact and reliance on fossil fuels. Polymers can undergo structural and chain changes due to photodegradation, oxidation, and hydrolysis. In most cases, enzymes and chemical reactions are involved in the biodegradation of organic matter. Based on their raw materials and production methods, biodegradable polymers fall into distinct categories. The three primary methods for their production are microbial fermentation, direct biomass extraction, and synthetic synthesis using biomass or petrochemicals. Biodegradable packaging materials can be derived from microbes, animals, and plants. In the presence of organic chemical or biochemical processes, these materials will decompose rapidly once released into the environment [25]. Moreover, they must be produced economically and sustainably. Proteins and polysaccharides are just two of the numerous food-grade materials used to create biodegradable packaging. The various types of biodegradable film are depicted in Figure 2 and its application in food products (Table 1).

#### 2.2.1. Proteins

Proteins are an abundant source of the most valuable polymers due to their superior gas barrier properties. Each amino acid can exist in various forms, and each protein contains 20 monomers of amino acids. As a result of the multilayer structure of proteins, different amino acid types with distinct enthalpies interact and bind at distinct sites [26]. Proteins are frequently used in biodegradable films due to their numerous beneficial properties, low cost, high nutritional value, and ability to form solid films [27]. The protein structure can be improved by a variety of physicochemical processes, including mechanical processing, heat, radiation, pressure, lipid interfaces, metal ions, acids, and alkalis [28]. Additionally, proteins have beneficial qualities that could make them a component in the creation of edible films [29]. Consuming foods wrapped in edible films is an excellent way to preserve nutrients and flavor by controlling the amount of oxygen, carbon monoxide, and ethylene that enters and leaves the food. Moreover, nitrogen supplies for those who can serve as fertilizers can be provided during the decomposition of protein-based films [30].

Casein and whey proteins have been demonstrated to be advantageous for the production of biodegradable packaging materials. Caseins, which account for approximately 80% of milk’s protein, derive the majority of their functional properties from their ability to aggregate near their isoelectric point. The most common form of these proteins used in the food industry is sodium or calcium caseinate, which is produced by reacting casein solutions with sodium hydroxide or calcium hydroxide, respectively [31]. Caseinate has been utilized to produce mechanically and aesthetically superior films [32]. Approximately 20% of the proteins in milk are whey proteins, which include lactoglobulin, lactalbumin, bovine serum albumin, and immunoglobulins [31]. These consist of immunoglobulins, lactoglobulin, lactalbumin, and bovine serum albumin [31]. These globular proteins form incredibly cohesive films because of their exceptional gelling abilities. According to Azevedo et al. [33], the film made up of whey protein exhibits desirable mechanical and oxygen-barrier properties at inferior and moderate relative humidity. However, these films lacked an effective water vapor barrier, limiting their applicability as food packaging. The precise control of the whey proteins’ denaturation, association, and crosslinking is necessary in order to produce films with the desired functional characteristics [34,35]. Milk-protein-based films have numerous desirable characteristics, including softness, smoothness, tastelessness, and transparency. In addition, antimicrobial and antioxidant properties can be imparted to a product by encapsulating functional additives [36]. In the food industry, this packaging’s lack of durability and susceptibility to moisture are significant drawbacks.

Gelatin is a popular component of time-degradable films because it is a meat protein. Collagen is extracted by processing animal bones, skin, tendons, and hooves [37]. According to Rakhmanova et al. [38], gelatin is produced by heating collagen in a concentrated pH solution. This method purifies gelatin to be utilized as a dietary supplement and for other purposes. In order to create a gelatin gel, the solution needs to be heated above the temperature at which the coil-to-helix transition occurs [39,40]. This encourages the formation of crosslinks and increases protein concentration. Gelatin can be used to create thin films with desirable mechanical characteristics, which makes them a viable option for food packaging. However, their poor barrier qualities, particularly against water vapor transport, frequently limit their usefulness [41].

Plant proteins sequestered from zein, gluten, soybeans, nuts, peas, and sunflower are just some of those found in gelatin that can be used to make films that break down naturally [42]. As Vahedikia et al. [43] point out, zein, a corn protein that does not dissolve in water but concentrated alcohol solutions, is essential to producing edible films. There have been prior experiences using zein materials for food packaging [44]. According to Visakh [45], the proteins isolated from soybeans can also be used to create edible films, which are frequently created through baking or casting techniques [46]. Despite their stretchability and good mechanical properties, soy proteins rarely function as water barriers [47]. However, soy proteins allow for the development of smooth and stretchable edible films. Soybean films can be treated with hydrophobic additives to improve their water barrier properties. Nonetheless, this treatment may alter the films’ optical and mechanical characteristics [48]. Mohareb and Mittal [49] found that soy films performed better when supplemented with glycerol, gellan gum, or carrageenan.

Crizel et al. [50] examined the viability of preserving lard with a gelatin film containing papaya peel microparticles. The results demonstrated that incorporating microparticles significantly improved the gelation film’s mechanical properties. The efficiency of a film containing microparticles is due to the increased protein–protein interaction and the increased particle dispersion within the matrix. The gelatin film reveals well in several ways and is suitable for packaging applications compared with low-density polyethylene film. The prepared film was evaluated and found to be suitable for packaging applications due to its superior properties that were on par with those of low-density polyethylene-based food packaging. Other research found 60% shellac and 40% gelatin in the edible outer layer coating. It showed that bananas could be fresher for longer if coated in an edible film [51].

Oliveira et al. [52] discussed the use of cashew gum and gelatin films in the production of washing powder packets, solubilized films, and agricultural encapsulants. In order to increase its barrier properties, the gelatin concentration was increased. Conversely, the synergistic effect showed the most extraordinary adaptability of any of the materials tested. The film was only slightly less stable at high temperatures than gelatin film. These examples prove the value of incorporating gelatin film and cashew gum into final products to increase biodegradability. Huang et al. [53] evaluated the effect of electron beam irradiation on the antioxidant levels of bamboo leaf and fish gelatin-based films in 2020. After being exposed to radiation, molecules interacted more strongly with one another and crosslinked, reducing the flexibility of the polymer chains. This particular instance can be attributed to the increased relative humidity. After being exposed to radiation, bonds form, increasing the film’s surface’s polarity and decreasing the contact angle. Radiation-induced interactions within molecules are responsible for the outstanding thermal stability of the developed film.

#### 2.2.2. Polysaccharides

Polysaccharides in nature are abundant and degrade rapidly in the environment [54]. The term “polysaccharide” refers to various structures derived from plant and animal sources. Glucomannan, xanthan, agar, carrageenans, pectins, algins/alginates, gellans, curdlan, dextrans, levans, arabinoxylans, and pullulan are widely utilized in the food industry and other industries [55,56]. These polysaccharides can serve a variety of functions and display a variety of structures. They differ in size, amylose content, molecular confirmation, glycosidic bonds, and the role of various functional groups. Additionally, these polysaccharides have a high tendency for film formation. According to Zhu et al. [54] and Mostafavi and Zaeim [57], they have indeed been produced into films and coatings for preservation, including the preservation of meat, fish, fruit, and vegetables. Polysaccharides that create biodegradable films include starch, cellulose, chitin, chitosan, and hydrocolloid gums [58]. The physicochemical and functional properties of polysaccharide-based packaging materials vary based on molecular features.

Starch is used widely because of its low cost, high availability, biodegradability, and high renewability [59]. According to Shirazani et al. [60], polysaccharides’ molecules rearrange into crystalline-rich regions to form starch granules. The use of edible films made entirely of starch is limited, according to Ilyas et al. [61], because of their high water vapor permeability and subpar mechanical qualities. In order to enhance their functional performance, researchers have studied the effects of including additional additives. For instance, starch and polyvinyl alcohol were combined to produce a film with good water barrier properties, increasing its potential for use in commercial food packaging [62].

Cellulose is a polysaccharide source that is frequently employed in the production of biodegradable packaging. This material is the most abundant polysaccharide in nature [63]. It is derived from the hydrolysis of a wide range of plant species. Moghimi et al. [64] and Alizadeh-Sani et al. [58] are researching the potential use of cellulose derivatives as biodegradable film materials. Several examples are carboxymethyl cellulose, hydroxypropyl methylcellulose, and methylcellulose. According to recent research [65], there has been a limited adoption of cellulose-based films for use in food products due to their poor water vapor retention properties. Shanmugam et al. [66] compared the properties of nanocellulose and recycled nanocellulose films using a spray coating technique. Despite the RNC film’s roughly 50% greater air permeability than virgin film, it was deemed superior air resistance when used for packaging. Compared to the control film, the biodegradable film’s water vapor permeability increased by 97.7%, while its air permeability decreased. Agglomeration during recycling was linked to the film’s low barrier quality. For this reason, a homogenizer is advised for use in cellulose nanofiber solutions to prevent the formation of agglomerates. In conclusion, the application of RNC films was comparable to conventional packaging materials such as polyvinyl chloride, polyethylene, and polystyrene.

Achaby et al. [67] derived cellulose nanocrystals (CNCs) from sugarcane bagasse fibers. To evaluate its properties, they incorporated them into a polyvinyl alcohol (PVA) and carboxymethyl cellulose (CMC) composite film. Comparing the bionanocomposite with 5 wt % CNC to the PVA/CMC film, the TS and modulus of the bionanocomposite with 5 wt % CNC showed an optimal increase. These results indicated that the contact between the CNC and the matrix and the CNC’s high aspect ratio was essential for forming hydrogen bonds, which increased the strength of the bionanocomposites. It is possible that the increase in the barrier properties of bionanocomposite films can be attributed to the uniform distribution of CNC in the matrix, which lowers the numeral of voids in the biodegradable films. In light of these encouraging findings, biodegradable films have the potential to replace conventional food packaging materials.

When cellulose is the most plentiful amylose and amylopectin in nature, chitin comes next as the second most abundant polysaccharide. Chitin can also be derived to form chitosan, the deacylated form of the former compound [25]. Both of these compounds were excellent in forming biodegradable films that can be used widely in food products to extend their shelf life, especially fresh agricultural products such as fruits and vegetables [68]. Compared to chitin-based films, chitin-based films are weaker and more permeable to gas and moisture than chitosan-based films. However, their strength can be further enhanced by including other functional components or additives to form a blend of composite films [58,69]. One of the reasons why chitin and chitosan-based films are sought is their antimicrobial properties, which could extensively elongate the shelf life of the foods we consume [25,58].

Francisco et al. [70] produced biodegradable films from acetylated cassava starch and hydroxyethyl cellulose to preserve guavas (*Psidium guajava*, L.). Films with a higher concentration of hydroxyethyl cellulose were more precise and hygroscopic. Guava coated with the developed film exhibited increased firmness, preserved green skin color, and slowed ripening for 13 days. Consequently, this demonstrates that biodegradable film can extend shelf life while reducing environmental impacts. In addition, the films were found to be thicker, more translucent, and more hygroscopic, with a reduction in mass loss, increased rigidity, and they preserved the green color of guava skin. The use of edible hydrocolloid gums as a biodegradable packaging material is becoming a trend in the food industry. One commonly used material is pectin, which is usually obtained from various sources ranging from sugar beets to apples. Commercially, pectin is used as a thickener, a gelling agent, and a stabilizer in all sorts of products in the food industry [71]. However, recent studies stated that pectin-based films have extreme mechanical resistance and a low permeability to gaseous diffusion. However, they are said to have very high water vapor permeability, which is unsuitable for food products [72]. Because of that, pectin-based films were proven effective against food products with low water activity [73]. Overall, most hydrocolloid gums, including alginate, carrageenan, and agar, could be used to create biodegradable films to be applied in food packaging due to their intermolecular cross linking abilities [74,75].

#### 2.2.3. Lipids

Films made from lipids are widely used as coatings despite their relative rigidity. Many lipids are employed as biodegradable films, such as monoacylglycerols, diacylglycerols, triacylglycerols, phospholipids, free fatty acids, and waxes [76,77]. The benefits listed above are only the tip of the iceberg regarding the numerous advantages of using lipid-based films. It has been documented that films made from palm fruit oil are transparent and highly water-resistant [77,78,79,80]. Furthermore, Vargas et al. [81] found that coating them with sunflower oil films reduced the permeation of oxygen and water vapor to enhance food quality. The effectiveness of methylcellulose and chitosan films was enhanced using functional additives made from citrus fruit peel essential oils (EOs) [82]. This includes using citrus fruits such as lemons, mandarins, and oranges. Syafiq et al. [83] found that apples were preserved with essential oils from cinnamon sticks, allspice berries, and clove buds for up to a year after harvest. Homogenizing lipids typically prepare an oil-in-water emulsion with only an aqueous solution containing an emulsifier before their incorporation into biopolymer-based films. Mechanical, optical, and other functional characteristics of lipid films are influenced by their composition, size, concentration, and interfacial properties, so these factors must be tailored for each application [80]. According to Haq et al. [84], gum cordia can be mixed with lipids to change the film’s characteristics. Beeswax was added to the original gum cordia film formula to increase its versatility.

Beeswax was added to the films, which caused a reduction in their tensile strength, Young’s modulus, and elongation. Films that included beeswax had significantly lower water vapor permeability compared to films that did not. It was discovered that the activation energy of beeswax in films containing beeswax was more significant than in films without beeswax. Figure 3 shows the example biodegradable packaging application.

**Table 1 molecules-27-05604-t001:** Application of biodegradable film in food products.

Food Products	Biopolymeric Matrix	Coating Techniques	Results	References
-	Gelatin + papaya peel powder	Biodegradable film	Adding microparticles to gelatin film improved the mechanical properties Increased protein–protein interaction in a film containing microparticles and increased particle dispersion within the matrix results in efficiency	[50]
Fruits and vegetables	Shellac + gelatin	Biodegradable film	Higher concentration of gelatin (30, 40, 50%) exhibit reasonable stability	[51]
-	Cashew gum + gelatin	Biodegradable film	The combination of cashew gum and gelatin permits the formation of a biodegradable film The amount of gelation from 2.5 g to 7.5 g in the CG/G film blend significantly reduces water permeability	[52]
-	Cellulose nanocrystal (CNC) + polyvinyl alcohol (PVA) + carboxymethyl cellulose (CMC)	Biodegradable film	Enhanced tensile modulus and tensile strength Reduced water vapor permeability	[67]
Guava	Acetylated cassava starch (ACS) + hydroxyethyl cellulose (HEC)	Biodegradable film	The films with higher HEC concentration were more transparent and hygroscopic Guava coated with 75% HEC and 25% ACS or 100% HEC films increased firmness, maintained green skin color and reduced ripeness	[70]
Fruits and vegetables	Gum cordia + lipids + beeswax + glycerol monosterate	Biodegradable film	Biopolymer-based films possess lower OP than synthetic films	[84]

### 2.3. Edible Packaging

Nowadays, edible films have become increasingly popular and are seen as cutting edge in the food industry. It is important to consider (i) the material from which the film will be made, (ii) the food type to which the film will be applied, and (iii) the method of application when deciding on the best edible film or coating. The best outcomes can be achieved by selecting the optimal combination of these factors. There is also interest in exploring the potential of composite edible films, coatings, and nanomaterials. The future use of nanomaterial-containing films and coatings on food products depends on specific statutory parameters [85].

Edible containers have been used for hundreds of years to keep food fresh and make them look better which can be seen in Figure 4. However, scientists have only recently recognized it as a possible replacement for non-biodegradable synthetic packaging [76]. Edible packaging is a cutting-edge and practical way to package and transport food. Edible packaging can be anything from a chocolate drizzle on a wafer to a whole cake. As a result of their ability to prevent moisture loss and slow the rate of dangerous chemical reactions, edible films have been shown to improve the safety and quality of a wide variety of processed and fresh foods [86]. The amount of edible food that goes to waste will consequently decrease. Edible films have been shown to increase the quality and safety of various fresh foods by reducing the rate at which moisture is lost and harmful chemical reactions occur [86]. The edible films have low permeability and mechanical properties compared to standard synthetic plastic films [87]. Over the past 30 years, researchers have strived to perfect a film that can be eaten and then laminated onto regular plastic films. We need a solid understanding of how these substances behave in the body after consumption if we want to develop nano-systems that can be used safely and legally in commercial products. Therefore, more study is needed into the application of nanotechnology in edible films and coatings. This section focuses on the numerous varieties of edible packaging materials and its application in food products (Table 2).

#### 2.3.1. Polysaccharide-Based Edible Films

One of the most common types of natural polymer used in edible film production is polysaccharides, such as chitin, alginate, pullulan, and chitosan [88,89]. Edible films based on polysaccharides have a good oxygen barrier in the presence of the well-ordered hydrogen-bonded network. Because of the hydrophilic properties, polysaccharide-based films are not quite as efficient as other moisture barriers. Furthermore, polysaccharide coatings are oil-free, transparent, and can lengthen the shelf life of a product without causing anaerobic conditions [79]. It is possible to create polysaccharide-based films using either a wet or dry process.

Cellulose, the most prevalent organic polymer found in nature, can be utilized to create safe films for human consumption. The essential structural component of a cell wall is a linear 1,4 glucose homopolysaccharide that can be modified by adding methyl, hydroxyl, or carboxyl groups, and derivatives are produced. Coatings and films composed of cellulose derivatives, including hydroxypropyl methylcellulose, hydroxypropyl cellulose, methylcellulose, and carboxymethylcellulose are often used in the food industry [90,91]. For example, confectionery food items coated with methylcellulose form a barrier that blocks the passage of oil or lipids. Films or coatings based on HPMC prevent oil absorption, making them useful for applications such as fried food packaging [92]. Fagundes et al. [93] investigated the impact of the films in cold storage; these composite edible films, which usually contain hydroxypropyl methylcellulose, beeswax, and different food preservatives with antimicrobial potential, such as sodium benzoate, sodium ethylparaben, and sodium methylparaben, protected the food. The authors discovered that cherry tomatoes packaged in materials containing sodium benzoate were better preserved in terms of firmness and weight loss.

Biodegradable films are produced using starch, a renewable and abundant resource that provides the ideal matrix. Amylopectin is a branched polymer made up of (1–4) and (1–6) glucose molecules, while amylose is a linear polymer made up of (1–4) glucose molecules. Starch is a versatile, oxygen-permeable, and water-soluble material that is perfect for film formation [89]. The high hydrophilicity of amylose makes it a poor vapor barrier. In order to improve the quality of starch-based edible films, modification of the starch is required [94]. Ceballos et al. [95] fabricated an edible film from cassava starch and yerba extract. Compression molding was used to create the film from the thread produced using a twin-screw extruder. Films produced by compression molding have the potential to be thicker and more flexible than solvent-cast films [96]. Therefore, pectin is primarily composed of galacturonic acid and its analogues. Polysaccharides are commonly derived from apple pomace and citrus peel [97]. The esterification of pectin with methanol affects its gelation and film-forming properties. Methyl pectin can be further subdivided into low-methyl pectin and high-methyl pectin categories based on the extent to which they have been esterified [98,99]. Because of its biodegradability, biocompatibility, and diverse physicochemical properties, pectin is widely used in edible films. Gorrasi and Bugatti [100] previously created novel active coatings using a pectin matrix and a modified layered double hydroxide from renewable resources. The fresh apricots were coated with the active composite plasticized with glycerol, proving the formulation effective in extending the fruit’s shelf life. In addition to glucuronic acid, alginates contain mannuronic acid, hyaluronic acid, and guluronic acid. These additional components help to determine the alginates’ molecular weight and other physical properties [101].

Alginate polysaccharides are typically extracted from brown seaweed. Since alginate can thicken, stabilize, form a film, and act as a suspending agent, it can be used in the food industry to create films. It has been reported by Senturk et al. [102] that improving the physical properties of an alginate-based edible film by adding divalent cations as a gelling agent improves things such as moisture and color retention. Because of their inherent hydrophilicity, edible films and coatings will be more susceptible to moisture damage. Edible packaging made of sodium alginate and incorporating essential oils have been found to be an effective method of preserving homemade cheese, as determined by Mahcene et al. [103]. Pullulan is used as a thickener in the production of edible films, and coatings made with pullulan help keep fruits fresh for longer. Structure-wise, maltotriose units make up the bulk of pullulan. The fungus, *Auerobasidium pullulans* secretes the pullulan polysaccharide to protect itself from desiccation and predation [104]. When pullulan is fortified with glutathione and chito oligosaccharide, it acquires properties that make it more suitable as a food coating [76]. Pullulan’s inherent hydrophilicity reduces the effectiveness of the material as a water barrier and mechanical component. These limitations can be overcome by adding lipids and fatty acids such as beeswax, palmitic acid, and oleic acid [105].

The cell walls of phytoplankton and invertebrates are primarily composed of chitin. After being exposed to an alkaline solution, chitin is transformed into chitosan. N-acetylglucosamine, a disaccharide derived from glucose, serves as the structural backbone of chitin polysaccharide. Chitosan films can be used as a barrier against oxygen and carbon dioxide loss while also being edible. Kumar et al. [106] found that the physicochemical properties of chitosan-based edible films and coatings varied with the degree of chitin deacetylation. Chitin has many desirable properties, including biodegradability, biocompatibility, antibacterial activity, and low immunogenicity [107]. The use of transparent films made from chitosan has prolonged the shelf life of perishable foods and improved their safety. Natural gums are highly viscous and contain an antimicrobial component [108,109].

#### 2.3.2. Protein-Based Edible Films

Packaging food often involves using plant-based proteins such as those found in corn zein, wheat gluten, peanuts, quinoa, sesame seeds, milk, and soy. Animal-based film formers include, in contrast, keratin, casein, gelatin, egg white protein, myofibrillar protein, collagen, and milk whey protein [110]. In terms of nutritional value, protein-based edible films appear to have the most promise among the various types of edible films [111]. Edible films made from proteins have low moisture barrier properties and high mechanical and gas barrier capabilities. Protein-based films outperform lipid- or polysaccharide-based films in a number of important respects. This is because of the orderly arrangement of their hydrogen bonds, which results in desirable physical properties and gas-blocking effects [112]. Oxidation is a major factor in the degradation of lipid materials, both in terms of quality and longevity. Protein-based packaging that limits oxygen diffusion has practical applications in some situations [113]. The structure of a protein also plays a major role in determining how permeable it is to oxygen. The globular structural proteins in corn zein, wheat gluten, soy protein, and whey protein allow more oxygen to permeate edible films made from these ingredients [114]. Edible films made of protein can be used to package food items that are otherwise difficult to transport and store, such as beans, nuts, and cashew nuts, in convenient single serving sizes. The protein-based film can be manufactured using either a wet or dry process [115]. Since lactose causes crystallization, whey and casein proteins are preferable to total milk protein when making edible films [116]. Various edible films can be made by combining whey protein fractions, whey protein isolates, and whey protein concentrates with various emulsifiers and plasticizers [117,118]. A whey protein film has superior oxygen, aroma, and oil barrier properties in dry to slightly damp conditions. Furthermore, the film has the right properties for use in coating foods, separating layers of food, and making pouches. Scientists have recently studied the effects of adding probiotics and prebiotics to whey protein-based films on the films’ functional properties [119,120].

### 2.4. Smart Packaging

The term “smart packaging” describes a type of packaging that uses technology to react in a predetermined way to changes in the system, such as the quality, safety, or maturity of the food inside [58]. It helps to extend the shelf life of food, monitors whether or not it goes bad in transit, and provides information about the date and location of packaging [121]. According to Vanderroost et al. [122], intelligent packaging, which can report on the freshness and condition of its contents in real time, improves supply chain security and efficiency. In addition, smart packaging enhances the “Hazard Analysis and Critical Control Point” (HACCP) and “Quality Analysis and Critical Control Point” (QACCP) processes, which aim to control, detect, prevent, reduce, and eliminate any potential problem that could affect the product and its final quality [123]. Indicators, sensors, and data carriers (Figure 5) are the backbone of a smart packaging system, which can be implemented at any point in the packaging life cycle, from manufacturing to recycling [124]. Sensors and indicators provide information on product quality, whereas data carriers concentrate on supply chain logistics [125].

#### Indicators

The purpose of indicators is to inform consumers about the presence of a substance, the reaction between two or more substances, or the concentration of a substance [126]. By analyzing color changes, color intensity changes, and color diffusion along a path, indicators can collect information about packaged foods [127]. Based on this fundamental principle, indicators demonstrate an irreversible change in color or intensity [122]. There are three main types of smart packaging indicators: freshness indicators, time–temperature indicators, and gas indicators, which may be external or internal depending on their location on the packaging [128].

A freshness indicator is a smart device that monitors food quality during transport or storage. Exposure to harmful conditions and exceeding the recommended shelf life are common causes of deteriorating freshness. The freshness indicator can provide quick information about a product’s quality by identifying the chemical processes that result in food spoilage due to microbes [128]. Biogenic amines, volatile nitrogen compounds, glucose, organic acids, ethanol, carbon dioxide, and sulfur derivatives are all examples of metabolites known to be produced during microbial growth and thus serve as potential freshness indicators, as stated by Poyatos-Racionero et al. [129]. Synthetic and natural dyes that change color when exposed to acidic conditions are used in most freshness indicators [126]. For their consistent color shift across a wide pH range, anthocyanins are the most widely used natural pigments in smart packaging [58]. The anthocyanin flavylium cation dominates at acidic pH levels, giving the pigment a predominately red color. The anthocyanin structure is hydrolyzed by the flavylium cation in slightly acidic and neutral environments, yielding the carbinol pseudo-base and the quinoldal bases, respectively. Anthocyanins are chemically unstable and degrade into green or yellow chalcone species when exposed to strong alkaline conditions [130]. As a result of their sensitivity to pH changes, anthocyanins can be used as pH-indicators over a broad pH range [131] (Table 3).

Park et al. [132] developed a chitosan-based edible intelligent film that included oil extracted from clove buds and red cabbage to indicate fish freshness. During the growth of fish-spoiling bacteria, the indicator’s color transforms from violet to dark blue. In addition, an intelligent film containing anthocyanin derived from roselle and starch was created for potential applications in monitoring the freshness of meat. Highly sensitive to changes in pH, the indicator film turns from red to yellowish green in an alkaline environment [133]. In addition, curcumin biodegradable films incorporating modified rice starch were developed by Erna et al. [134] for the detection of hypoxanthine in poultry and fish. He et al. [135] developed a carrageenan and gelatin-based edible film that qualitatively evaluated the freshness of grass carp fillets. In light of these findings, the packaging was altered to change colors (from yellow to red) to alert consumers to an impending expiration date.

In addition, Lu et al. [136] created a hydrogel of nanocellulose derived from sugarcane bagasse for detecting spoiled chicken breast. This indicator included a poly (ether-block-amide) film on the inside, which was followed by a layer of eight color-changing polymer-immobilized pH dyes, and finally a poly(ethylene terephthalate) film on the outside. The indicator hydrogel’s optical color switched from green to red as the log CFU/g went above the threshold that is considered safe for human consumption and complex, volatile emissions. Additionally, a sophisticated carboxymethyl cellulose, cellulose nanofiber, and shikonin-based pH-sensitive indicator system was created to track the quality of seafood products [137]. Similarly, Baek et al. [138] created a pH-sensitive dye-based freshness indicator system that enables the monitoring of observable color changes brought on by the production of carbon dioxide and volatile acids during kimchi storage. Kuswandi et al. [139] used red cabbage anthocyanins immobilized on a bacterial cellulose membrane to develop a pH sensor that could be integrated into smart packaging and used to track the expiration dates of food products. Similarly, Taghinia et al. [140] developed and detailed a smart packaging system for monitoring shrimp freshness using *Lallemantia iberica* seed gum and curcumin.

Fruits and vegetables undergo ripening after harvest because of an increase in ethylene levels produced by the breakdown of plant tissues. The simple phytohormone ethylene triggers the senescence and browning of chlorophyll. Indeed, it is the single most important factor in the spoilage of perishable goods [141]. The susceptibility of packaged foods to chemical processes of decay, such as oxidation, can also be affected by the presence or absence of gases such as oxygen. Some of the gases are also produced by bacteria that have contaminated the product, making them useful indicators of the quality and safety of the food product. As a result, developing intelligent biodegradable films capable of detecting and identifying specific gases is crucial [142]. Gas indicators are also commonly used to test the efficiency of active packaging components such as oxygen and carbon dioxide scavengers or to locate packaging leaks [124]. According to Lamba and Garg [143], the gases contained in a package’s headspace can be affected by the food’s activity level, the package’s composition, and the environment in which the package is placed. Gas indicators, which can be labels or writing on the packaging materials, can be placed either inside or outside the packaging to monitor changes in gas composition, providing a means of ensuring the quality and safety of the food contained therein [121].

Recently, scientists have been looking into the potential of anthocyanins as natural dyes in combination with natural polymer matrices to create non-harmful, biodegradable colorimetric indicators for use in the food industry [144]. Adhesive labels, printed layers, or film embedding are just some of the methods proposed for incorporating colorimetric gas indicators into packaging materials, which can provide results more quickly and at lower cost than an analytical instrument [145]. In food packaging, oxygen is removed and replaced with nitrogen or oxygen scavengers to prevent microbiological and biochemical deterioration. Nonetheless, poor packaging, defects, or damage may allow oxygen to enter the package during transport and storage. Visible oxygen indicators are required to quickly and easily confirm the presence of oxygen within a package without the need for specialized equipment or laboratory investigation [146]. Previously, Jang et al. [147] proposed a method for measuring the oxygen concentration in food using naturally occurring organic compounds such as cysteine and laccase, where the rate of color change correlates with the oxygen concentration.

Some natural pigments undergo chemical changes when exposed to specific gases, allowing them to be used to create gas sensors. Chen et al. [148] developed a colorimetric gas indicator to monitor changes in the carbon dioxide content of fresh green bell peppers. As the colorimetric films degraded following the slicing of the bell peppers, they shifted from green to orange. Abu-Hani et al. [149] introduced a novel gas sensor based on chitosan film with engineered conductivity in a recent study, adjusting the chitosan film’s conductivity by combining it with glycerol ionic liquid. Moreover, Ko et al. [150] developed a biodegradable, silicon-based, flexible electronic system for detecting nitrogen oxide species. This system has record-breaking response times of 30 s and recovery times of 60 s, with a sensitivity of 136 Rs. Zhai et al. [151] discovered that a starch/polyvinyl alcohol film loaded with anthocyanins from roselle was an effective tool for determining the freshness of silver carp. The colorimetric label demonstrates that the transformation from purple to yellowish-green was caused by the production of volatile primary nitrogen amines. Ma et al. [152] used Tara gum/polyvinyl alcohol film containing curcumin as a smart colorimetric packaging material to detect the production of ammonia.

Time–temperature indicators (TTIs) are useful for keeping an eye on perishables such as fruits and vegetables to make sure they do not lose any of their quality during storage. Perishable foods must be carefully monitored and managed for temperature throughout the entire distribution chain to ensure their safety and freshness. Significant amounts of food are lost or discarded because of poor or unreliable temperature management at various stages of the food supply chain [126]. TTIs are widely used to monitor the freshness and safety of packaged foods and determine if they have been damaged by excessive heat during transit or storage [153]. Yousefi et al. [154] detail the development of a TTI-containing smart packaging material that employs natural food pigments as indicators to monitor and verify the integrity of perishable foods such as fruits and vegetables. Indicators of partial history alert buyers when temperatures above a certain threshold suggest that microorganisms may have survived the freezing and defrosting process or that proteins may have been denatured. However, a full history indicator monitors food storage temperatures and provides updates throughout the product’s entire shelf life [124].

Typically, a temperature sensor’s activation energy must be exceeded prior to a change in state, such as a change in color. According to Taoukis et al. [155], temperature sensors should have activation energies between 10 and 40 kcal/mol. The shelf life of food can be predicted with reasonable accuracy using a well-designed temperature sensor, as discovered by Zhang et al. [156]. According to Goransson et al. [157], TTIs are widely used in the food packaging industry because they are inexpensive to manufacture and easy for consumers to read. Methods for detecting temperature, such as diffusion, polymerization, microbial growth, enzymatic reaction, thermochromic reaction, photochromic reaction, electronic, and surface plasmon resonance, are grouped into distinct categories for TTIs [156]. Temperature-sensitive sensors are also classified according to how they work;

Critical temperature indicators (CTI), which tell you if food has been heated above or cooled below a specific temperature during its lifetime;Critical temperature/time integrators (CTTI), which tell you if food has been heated above or cooled below a specific temperature for a more extended period; andTemperature sensors describe whether the food has been heated above or cooled below.

The intelligent packaging developed by Maciel et al. [158] involves the incorporation of a temperature-sensitive anthocyanin into a chitosan/cellulose matrix, causing the matrix to change color from violet to yellow between 40 and 70 °C. Saenjaiban et al. [159] recently developed TTIs using glycerol and carboxymethyl cellulose-encapsulated polydiacetylene/silver nanoparticles and silver nanoparticles. Given that silver nanoparticles have a higher thermal conductivity and polydiacetylene has a larger exposed surface area, the films containing polydiacetylene and silver nanoparticles undergo color changes from purplish-blue to purple and from purple to reddish-purple over time. The degree to which these indicators’ colors shift depends on the time–temperature profile to which the packaged food has been subjected. Therefore, these TTIs can infer whether a food product likely went bad during storage. The ability of a temperature sensor to react to changes in its operating temperature is essential as shown in Figure 6 [153].

### 2.5. Active Packaging

Recently, “active packaging” has piqued the interest of scientists who work in the food packaging industry. Adding “active” substances to packaging that modify the food’s metabolism, increase its resistance to damage, boost its quality, and lengthen its shelf life is one way to achieve these goals [160]. Active packaging is a novel way to preserve or extend the shelf life of food goods while maintaining their quality, safety, and integrity. According to European regulation (EC) No 450/2009, “active packaging” is “packaging systems that interact with the food by deliberately incorporating components that would release or absorb substances into or from the packaged food or the environment surrounding the food” [161]. As shown in Figure 7, active packing systems can either act as scavengers (absorbers) or a releaser (emitters). In contrast to the former method, which involves the elimination of compounds such as moisture, carbon dioxide, oxygen, ethylene, and odor, the latter method involves the introduction of compounds such as antimicrobial compounds, carbon dioxide, antioxidants, flavors, ethylene, and ethanol either into the packaged food or into the headspace [162].

Active packaging is a type of packaging in which the container, its contents, and the environment work together to extend the product’s shelf life [163]. Active ingredients are added to the packaging material in order to improve the quality and extend the shelf life of the packaged food by changing the environment in which the food is stored, altering its metabolism, and providing better protection [160]. Active packaging helps reduce food waste by extending the shelf life of perishable items without significantly affecting their taste or freshness. Although fresh produce is the most common application of active packaging, it can be used for any type of food [164]. Fresh produce has a high perishability and spoils quickly when stored improperly, which accounts for this.

Typically, active packaging materials were made with synthetic polymers derived from petroleum, which do not biodegrade. Food packaging is one source of non-biodegradable municipal solid waste, which accumulates in landfills and has negative effects on the environment because there is nothing left to do with it once it has been consumed. This paves the way for the use of plant-based biodegradable polymers in functional food packaging [165]. Recently, there has been a shift in active packaging toward using processes and components derived from natural sources in order to boost product efficacy, safety, organics, and sustainability. Essential oils, which are often used as natural additives in packaging applications due to their potent antioxidant and antibacterial properties [166], are an excellent example of a natural source that can be integrated into a product. Therefore, advancements in active packaging solutions with enhanced preservation properties have been accomplished, which can benefit the food and packaging industries by decreasing food waste and making waste management more straightforward.

Multiple analyses have highlighted the potential of active packaging technology to provide shoppers with safer, “healthier”, and better quality products [167], leading to the development of a wide variety of active packaging systems. It is possible that the theoretical results of active packaging technologies will not be replicated in practical settings involving human food. The intricate composition of the food may affect the performance of the packaging. For example, the rates of release, absorption, and diffusion of the active substance can be changed. In addition, active compounds or carriers might react with food components or bind to them, which would reduce the effect. It is important, then, to look at active packaging studies for specific foods with a critical eye. This will help food and packaging experts understand the benefits of these systems better and may speed up their use in commercial settings.

#### 2.5.1. Active Scavenging System (Absorber)

Using oxygen scavengers, which aim to remove excess oxygen from the packaging or increase boundary qualities by acting as an active protective layer [168,169], is one of the most important methods of active packaging. The food industry works to keep oxygen out of food packaging because some foods are sensitive to it. This is typically accomplished through gas flushing or MAP procedures. Nevertheless, the concentration of residual oxygen in the package primary focus between 0.5 and 5% and may rise during storage [170]. The oxygen dissolved in the food itself may be released into the headspace of the package to reach equilibrium with the gas phase, oxygen permeating through the packaging material or poor sealing, depending on the circumstances [171]. By inducing oxidation [125] or promoting the growth of aerobic bacteria [172], oxygen in packaging reduces the quality and shelf life of certain foods, resulting in color changes [173], sensory changes [174], and nutritional losses [175]. Food packaging with a lower residual oxygen content is possible with the help of oxygen scavengers [176].

In Europe, unlike in Asia or the United States, sachet-based applications are not widely accepted by consumers since they are perceived as foreign bodies in food containers [177]. In reality, one of the downsides of such sachet-based active packaging technologies is the potential for unintentional breaking, which might result in involuntary intake of the content. Other disadvantages include the need for an additional packing process step or their incompatibility with liquids or moist meals due to moisture sensitivity [178]. Alternatively, various innovative oxygen-scavenging technologies have been developed over the past decade, such as integrating active chemicals directly into packing films or containers. However, only a few of these have been successfully applied in real-world food systems. As a result, research demonstrating the benefits of alternate oxygen-scavenging systems to specific food products is uncommon.

Recently, Lee et al. [179] developed different formulations of nonferrous oxygen scavengers involving activated carbon and sodium L-ascorbate in different ratios of components (1:1, 1:2, 1:1.4, 1:1.6, 1:1.8 and 1:2) to preserve the raw meatloaves. The optimized nonferrous OS of activated carbon and sodium L-ascorbate at a ratio of 1:1.6 was able to lower thiobarbituric acid reactive substances and microbiological changes of meatloaves. Meanwhile, Ramakanth et al. [180] designed a UV-activated OS system based on natural rubber latex from Hevea brasiliensis. The results showed that the designed UV-activated OS system can be an effective alternative to iron-based OS and is suitable for foods that are moderately to highly susceptible to oxidation. Furthermore, He et al. [181] also innovated mesoporous silica nanospheres that exhibited excellent oxygen scavenger performance. These examples show that oxygen scavengers can be made up of different substances such as iron, ascorbic acid, UV, and palladium, which provide different benefits that aim to reduce the percentage of oxygen in foods.

Ethylene (C_2_H_4_) is a plant growth regulator that promotes ripening and senescence by increasing the respiration rate of fresh and barely processed climacteric produce and lowering the shelf life during postharvest storage [182]. Ethylene also hastens chlorophyll degradation rates, particularly in leafy vegetables, and promotes fruit softening [183]. For these reasons, removing ethylene from the product environment through ethylene scavengers delays ripening and senescence, improving quality and extending shelf life. Wang and Ajji [184] invented a novel ethylene scavenger composed of pumice and potassium permanganate with smaller particle sizes and lower relative humidity that favored ethylene removal. Pumice acts as ethylene adsorbents while potassium permanganate oxidizes ethylene. Therefore, the ethylene scavenger’s mechanism has effectively extended avocado’s shelf life by one week. Meanwhile, Pirsa et al. [185] used a nanocomposite material as an ethylene absorbent for bananas. The developed nanocomposite film consisted of carboxymethylcellulose, nanofiber cellulose and potassium permanganate hydrogel. The film has effectively and efficiently maintained the packed bananas’ humidity and physical appearances. In addition, Jaimun and Sangsuwan [186] incorporated vanillin and ethylene adsorbents onto chitosan-coated paper to preserve the quality of Nam Dok Mai mango fruit. The coated paper contained varying concentrations of zeolite or activated carbon at 0%, 0.1%, 0.2% and 0.4%, *w/v* of the ethylene absorbers. The optimum formulation was vanillin–chitosan-coated paper containing 0.2% (*w/v*) of zeolite and has shown to have the greatest ethylene adsorption capacity. Therefore, the developed wrapper has improved and preserved the quality of mango fruit and provided the least severity index of anthracnose disease throughout storage.

Moisture content and water activity are significant elements influencing the quality and safety of many meals [187]. Many dry products, for example, are susceptible to humidity during storage, and even modest relative humidity levels inside the containers can cause considerable quality deterioration. Increased moisture makes products more susceptible to microbial deterioration and may result in texture and appearance changes, limiting shelf life [188]. Other products, such as fresh fish, meat, and fruits/vegetables, benefit from a controlled high relative humidity inside the container to prevent drying. Furthermore, some surplus liquid caused by drip loss is usual for fresh products such as fish and meat. Consumers see fluids in packaging as decreasing the beauty of a product, making it less desirable [189]. Moisture control strategies in packaging can be classified as moisture reduction (for example, by modified atmosphere packaging (MAP) by replacing humid air in the headspace with dry gas [190], or vacuum-packaging by removing humid air in the headspace) [191], moisture prevention (by barrier packaging) [192], and moisture elimination (by applying a desiccant/absorber) [193]. Only the second group can be called active, since moisture reduction and prevention are more passive techniques. Humidity levels inside packages can also be managed by using packing materials with a solid barrier to water vapor.

Active moisture scavengers are further classified into two types: relative humidity controllers that scavenge humidity in the headspace, such as desiccants, and moisture removers that absorb liquids. The latter might take the form of pads, sheets, or blankets and are typically placed beneath fresh food in various packaging systems (MAP, vacuum, skin pack, and so forth) [194]. They are used for high water activity items such as fish, meat, poultry, fruits, and vegetables (exceptionally cut products) [195]. Such pads are typically made of porous materials, polymers (PP or PE), foamed and perforated PS sheets, or cellulose, which are then mixed with superabsorbent polymers/minerals/salts (polyacrylate salts, carboxymethyl cellulose, starch copolymers, silica/silicates) [196].

#### 2.5.2. Active Releasing System (Emitter)

In recent years, there has been a rise in the development of antioxidant-releasing packaging for food applications. To avoid lipid oxidation, synthetic antioxidants such as butylated hydroxytoluene (BHT) and butylated hydroxyanisole (BHA) have been frequently utilized in food packaging [197]. Natural antioxidants such as polyphenols, tocopherols, plant extracts, and essential oils (EOs) are increasingly popular in active packaging materials. Subsequently, Pateiro et al. [198] innovated an antioxidant-active packaging using only green tea extract and oregano essential oil to evaluate the effectiveness of the packaging in sliced cooked ham. The combination of two powerful ingredients had effectively reduced the microbial growth and retained the original color of sliced cooked ham. Meanwhile, Wrona et al. [199] developed an antioxidant active packaging based on pure essential oils and vegetable oils to be tested on meat samples. It has been found that film with the presence of flaxseed oil can extend the shelf life of fresh meat by 22%. The optimum packaging corresponds to 50 μm LDPE film with antioxidant capabilities. Furthermore, a new and novel active coating was designed using Cucumis metuliferus fruit extract as the natural antioxidant additive to be implemented in food packaging applications [200]. Those findings showed that these natural coatings can be introduced to the typically used LDPE at the industrial level, as these types of packaging also have been enhanced in terms of their oxygen barrier properties without influencing the transparency of the final product.

Carbon dioxide is soluble in the aqueous and lipid phases of food items. The antibacterial action relies mainly on the solubility rate and amount of carbon dioxide dissolved in the food product [171]. Carbon dioxide solubility increases with decreasing temperature [201] and varies for different food products based on surface area, pH, and composition (water, fat, protein) [202]. The antibacterial action has been discovered to be proportional to the partial pressure of the gas [203]. In terms of food packaging, that means that the total amount of CO_2_ present in the package’s headspace is critical for the effect. There is the notion of a CO_2_-releasing device to be used in modified atmospheric packages to maintain high CO_2_ headspace levels during storage, allowing for reduced package volumes (lower gas to product (g/p, volume ratio) and longer shelf life) [182].

Incorporating CO_2_ emitters in redesigned atmosphere packages may allow for enhanced filling, smaller package sizes, improved transport efficiency, and a net reduction in environmental effect [204]. Carbon dioxide emissions from a tailored emitter system may also help to reduce packaging deformation by compensating for CO_2_ absorption into the food product during the first phases of storage [202]. In this manner, it prevents the production of negative pressure in modified atmosphere containers, which increases product drip loss and may give the packages an unappealing aspect to the consumer [205]. Furthermore, the inhibition of spoilage bacteria growth and the extension of shelf life for fresh food products at sustained high CO_2_ levels in the packages will have a knock-on effect in the form of reducing food waste. This issue is gaining increasing attention and priority in western parts. Emitters are often in the shape of a pad or sachet, and they are frequently paired with a liquid absorber. When the absorbent pad collects liquid seeping out of the product, the active substances inside the pad react, resulting in the production of CO_2_ [125]. The field of CO_2_ emitters has grown dramatically over the previous decade, as seen by increased research activity and the marketing of commercial CO_2_ emitters.

There has been increased activity in developing antioxidant-releasing packaging for food applications in recent years. Synthetic antioxidants, such as butylated hydroxytoluene (BHT) and butylated hydroxyanisole (BHA), have been widely used in food packaging to prevent lipid oxidation. There is growing interest in including natural antioxidants such as polyphenols, tocopherols, plant extracts, and essential oils (EOs) to active packaging materials. Antimicrobial food packaging is a system that is intended to prevent the growth of spoilage and pathogenic microorganisms [206]. Essential oils, enzymes and bacteriocins, antimicrobial polymers and organic acids, their derivatives and other organic compounds, and antimicrobial nanoparticles are the most researched active substances and materials utilized in antimicrobial food packaging systems.

Recently, Al-Moghazy et al. [207] designed a cellulose-based adhesive composite to serve as an active packaging material and fabricated allyl isothiocyanate (AIC) onto it. The adhesive composite is composed of gelatin electrospun fibers developed using an electrospinning apparatus. The designed product proved to have antimicrobial activities against *Staphylococcus aureus* ATCC 25923 and *Escherichia coli* O157:H7 and has proven to extend the lifespan of cheese by twice its natural lifespan, from 4 to 8 weeks. Priyadarshi et al. [160] incorporated citric acid and glycerol into chitosan film to create an active packaging for green chilli. The prepared film has shown to have better water resistance and moisture barriers, reducing the transmission and permeability of water vapor in and out of the green chilli. Therefore, the developed pouches effectively preserved the quality and appearance of green chilli.

Subsequently, Ashrafi et al. [208] developed a biocomposite film composed of chitosan and kombucha tea to extend the shelf life of minced beef meat. The developed film has antioxidant activity and provides a protective effect against ultraviolet (UV). Furthermore, the composite film had successfully retarded lipid oxidation and the microbial growth of minced beef during 4 days of storage, therefore, prolonging the shelf life of the meat by up to 3 days. Takma and Korel [209] congregated chitosan and alginate with black cumin oil onto polyester films to form a multilayer active packaging film to maintain chicken breast meats’ quality and shelf life. The assembled packaging film positively impacted the safety and quality of chicken breast meat via the antimicrobial action of the active packaging materials. Similarly, Alizadeh-Sani et al. [210] implemented active packaging to extend refrigerated meat’s shelf life. The research involved an active nanocomposite packaging composed of cellulose nanofiber, whey protein, titanium dioxide particles, and rosemary essential oil to improve the functionality of the packaging film. The designed packaging film has notably increased the shelf life of lamb meat from only around 6 days to 15 days by reducing the microbial growth, retarding lipid oxidation and lipolysis during storage.

Meanwhile, Kumar et al. [211] developed a novel bio-nanocomposite packaging film for preserving the quality and freshness of green grapes by using zinc oxide nanoparticles that were synthesized in an environmentally friendly manner using Mimusops elengi fruit extract and incorporated into an agar matrix. The packaging film showed a positive impact on the appearance and shelf life of green grapes, where the fruits were still looking fresh until day 14 to day 21. Green bell pepper was used to test a novel nanocomposite active film developed by Salama et al. [212] that included chitosan biguanidine, carboxymethyl cellulose, and titanium oxide nanoparticles. Green bell peppers that had been coated in nanocomposite films showed improved resistance to mass loss and deterioration when stored for long periods of time. The results indicate the generated nTiO_2_ nanocomposite films’ ability to extend food shelf life. Figure 8 and Table 4 shows the role of active packaging as quality control of fruits and vegetables.

## 3. Food Safety Issues in Food Packaging

Food and packaging can adversely affect the quality and safety of the food products. The food industry is one of the biggest industries that has been affected by the sustainability trend. Over the years, food packaging trends have evolved, but significant problems and obstacles remain. This consists of food waste, plastic pollution, and excessive resource consumption. Previously, Lee [215] stated that food packaging is designed to protect and preserve the food products from possible physical, chemical, microbiological, or other hazards that ultimately can impact their safety and quality. In particular, the interaction between food and its packaging is crucial, as food comes into contact with the packaging. Despite food packaging’s main goal of protecting food from environmental factors, food–packaging interactions can compromise food quality and safety [216].

Food contact materials are one of the most overlooked sources of chemical contamination in food as they may release contaminants and then transfer them into the food. Migration is influenced by physicochemical properties of the migrant, packaging material, and food composition such as fat content. In other cases, such as the use of polysaccharide coatings for agricultural produce will be regulated in precisely the same manner as other dietary components and considered as one of the important components for fruits and vegetables [217]. Polysaccharide-based coatings are mainly used in the production of edible packaging as it is classified as GRAS by the FDA and are commonly used in Good Manufacturing Practices (GMP). However, the manufacturer of those biopolymer-based coatings may request GRAS status if the biopolymer component used is not GRAS-approved by demonstrating the safety of the final products. Furthermore, the manufacturer may file a GRAS Affirmation Petition to FDA and they might be able to commercialize the products with approval from the FDA.

Moreover, active and intelligent packaging might incorporate colors, antioxidants, antimicrobials, and additional nutrients to optimize the functionality of the films or coatings. According to EU regulations, food additives in films and coatings must be identified and labeled on packaging with their functional category as well as their name or E-number. Most countries consider antimicrobials as food additives as their primary purpose is to extend the shelf life of food. The application of edible coatings and their concentrations are governed by national regulations. In general, all substances listed on the label must be accurately described because edible films and coatings have become an inseparable part of fresh food [218].

Other significant issues concerning allergenic components in edible films and coatings are addressed in regulatory statutes. Polysaccharide extracts frequently contain protein residues [79]. For example, guar gum extract may cause occupational asthma and allergies in some cases due to the presence of protein (<10%) [219,220]. Nonetheless, due to the small number of cases reported, guar gum is not a critical food allergy [221]. Aside from polysaccharides, edible films and coatings can be manufactured with allergen-rich proteins such as milk (casein, whey), wheat (gluten), soy, and peanut [222].

Currently, packaging companies prioritize eco-friendly materials and packaging solutions. By preventing chemical residues and extending shelf life, food packaging provides customers with simple food product safety, convenient handling, and transport. Numerous materials, including plastics, glass, metals, and papers and their composites, have been utilized for food packaging. The significance of harmful substances leaching from packaging materials into food is, however, of more significant concern due to the increased health consciousness of consumers.

## 4. Future Perspectives

Recently, researchers have focused on the feasibility of using biopolymers to fabricate films and coatings for consumable food. The flavor and freshness of the product can be maintained over time by using edible films and coatings. The outcomes of drizzling edible solutions containing additives on freshly sliced fruits, vegetables, and meats have been promising. Adding new antimicrobial, antioxidative, and antibrowning agents could improve food safety and quality. It is possible to acquire a deeper understanding of the synergistic effects of edible coatings and active agents. Combining film-forming biopolymers to improve the structure’s properties is another promising strategy. There is also a dearth of research on developing novel synergistic gelling systems. So, it is feasible to carry out such a comparative study. Most studies on edible coating applications have been conducted in the laboratory, resulting in a dearth of real-world applications. Future applied research should concentrate on edible coatings because they can increase the shelf life of food. To eliminate these issues, coating application methods can be modified to include a recycling process that does not waste an excessive amount of coating solution, reduce the number of microorganisms in the solution during recycling, develop spraying techniques for uneven surfaces, create industrial-sized vacuum tanks, etc.

This facilitates the widespread use of edible films and coatings in novel ways. It is anticipated that the development of edible films and coatings containing two or more nanomaterials will be a future trend in edible packaging. These products are anticipated to have better gas barrier qualities, superior product stability, distinctive colors and flavors, and higher nutritional value. In order to maintain food stability, new approaches for regulating mechanical properties and gas transport must be investigated. Edible films and coatings need to assist food products in adapting to their environments by altering their properties in response to environmental factors such as relative humidity and temperature. A method for effectively regulating the flow of oxygen, carbon dioxide, and water vapor in a system requires careful consideration of the nanomaterials.

## 5. Conclusions

Nowadays, coatings are receiving considerable attention due to the potential applications of nanotechnology, such as the addition of conductive or antimicrobial properties. There are several advantages to using natural-based films and coatings instead of adding preservatives directly to food. Food preservatives such as antibacterial and antioxidant agents are only allowed to be used in the appropriate amount as the packaging material to minimize the interaction between the chemical compounds and food product. Additionally, antimicrobial/antimicrobial films or coatings allowed a controlled release of those food preservatives, reducing the risk of reaction with other food components. Ideal edible coatings or films would help to prevent water loss due to evaporation, as they operate as the sacrificial moisture agent, minimize the loss of desirable odor and flavor volatiles due to evaporation, retard the growth of microorganisms due to respiration, and reduce the exchange of gasses due to gas exchange. Furthermore, the availability of natural-based films or coatings is high, and are most cost effective in terms of its manufacturing cost as compared to current packaging that incorporates the applications of nanotechnology.

Alterations to the atmosphere caused by the barrier must not encourage anaerobic respiration or the production of harmful volatiles. Prolonging the shelf life of freshly cut fruits and vegetables, meat, poultry, seafood, and cheese, edible films, and coatings is a viable option for accomplishing these goals, as shown by a review of the current literature on the topic. Researchers can use this synopsis to create or improvise the current functional coatings, as some of the current edible films or coatings are well-known for being hard and brittle. Extending the shelf life of a product through bio-packaging has been shown to reduce costs associated with spoilage due to natural ripening. The biomaterials and types of biologically active compounds used to coat products determine whether or not those products’ sensorial, physicochemical, or nutritional qualities are enhanced. Despite their potential usefulness in producing food-safe films and coatings, research into several biopolymers and additives that have such properties has lagged. This review might give some good ideas about the current methods for keeping food safe and fresh.

## Figures and Tables

**Figure 1 molecules-27-05604-f001:**
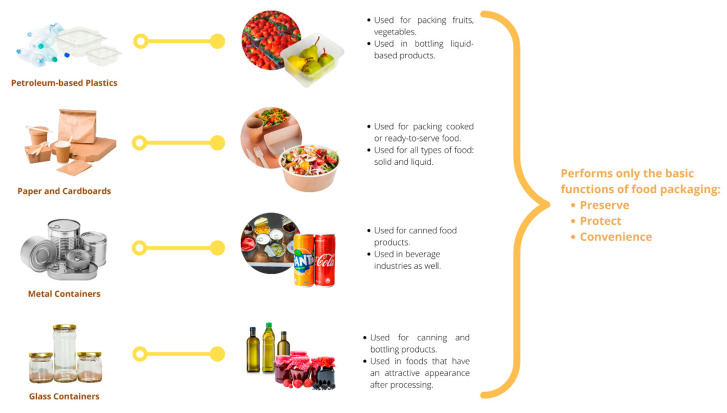
Application of conventional packaging.

**Figure 2 molecules-27-05604-f002:**
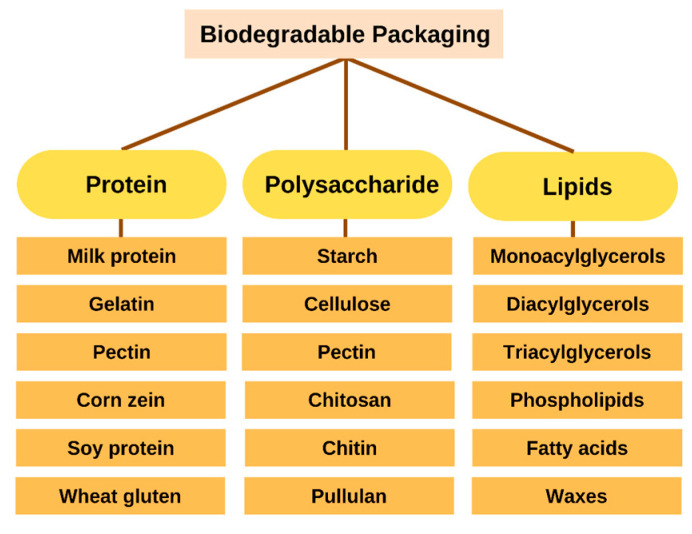
Classification of biodegradable packaging.

**Figure 3 molecules-27-05604-f003:**
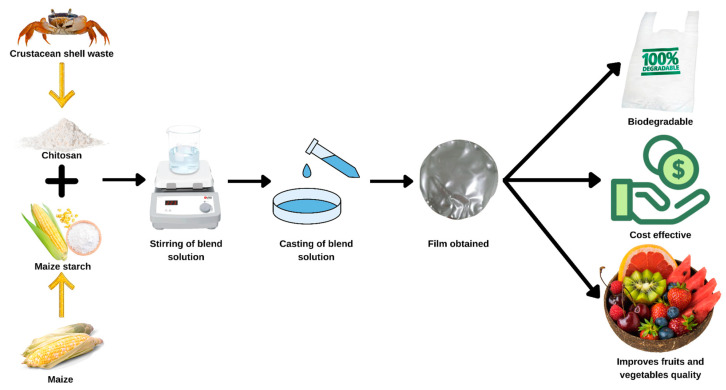
Illustration of biodegradable packaging.

**Figure 4 molecules-27-05604-f004:**
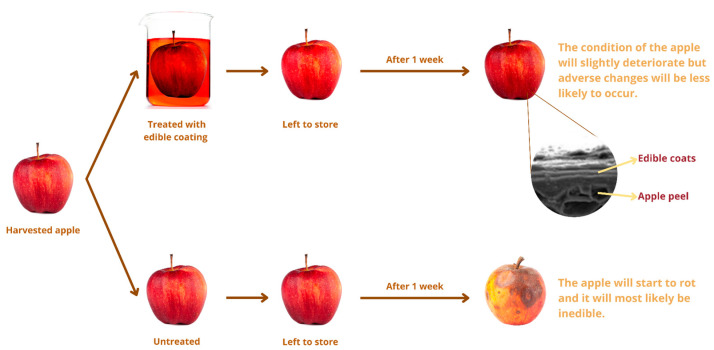
Application of edible coating.

**Figure 5 molecules-27-05604-f005:**
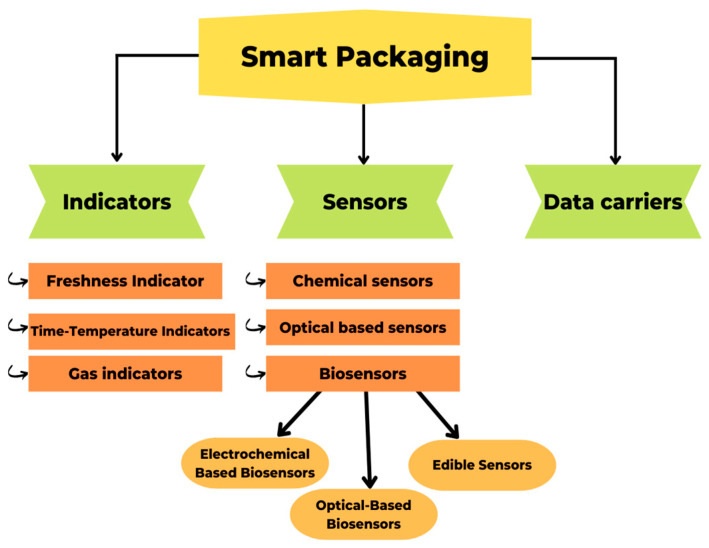
Types of smart packaging and its categories.

**Figure 6 molecules-27-05604-f006:**
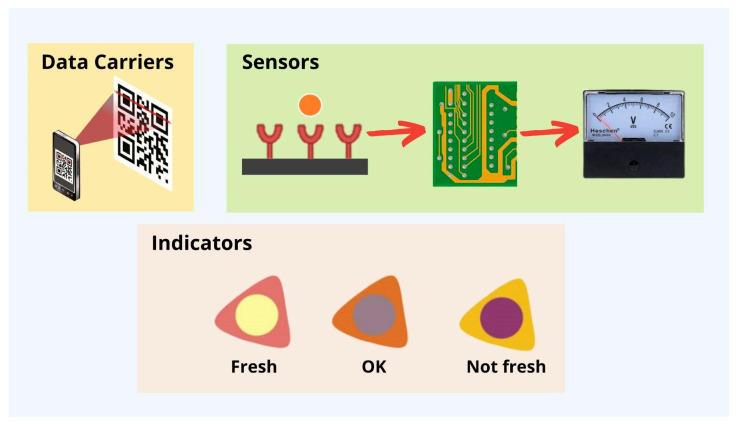
Typical intelligent packaging for food products.

**Figure 7 molecules-27-05604-f007:**
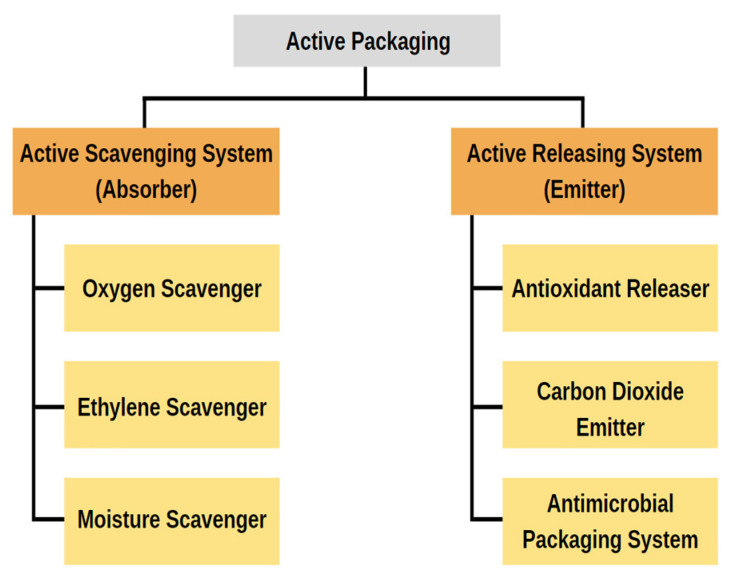
Types of active packaging and its categories.

**Figure 8 molecules-27-05604-f008:**
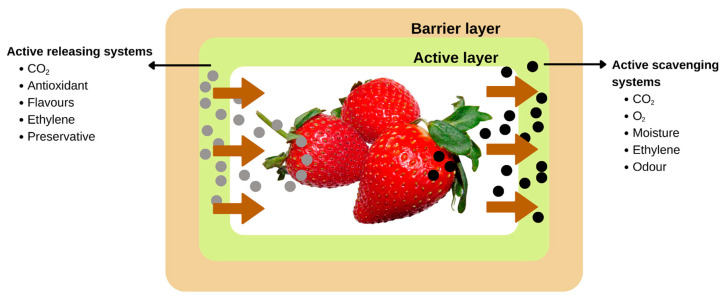
Active packaging as quality control of fruits and vegetables.

**Table 2 molecules-27-05604-t002:** Application of edible packaging in food products.

Food Products	Biopolymeric Matrix	Coating Techniques	Results	References
Cherry tomatoes	Hydroxypropyl methylcellulose	Immersion	Reduced weight loss, respiration rate, and preserved firmness of the cherry tomatoes	[93]
-	Cassava starch	-	Film with 10 wt % of native starch extract was more hydrophobic and tensile resistant	[95]
Apricot	Apples pectin + LDH-salicylate + glycerol	Dipping	Plasticized composites containing 4 vol% glycerol showed a better barrier property improvement	[100]
Homemade cheese	Sodium alginate + essential oil (*O. basilicum*, L.; *R. officinalis*, L.; *A. herba alba*; *Asso. M. pulegium*, L.)	Immersion	The growth of *Staphylococci Salmonella* for all types of cheese were completely inhibited	[103]
Homemade cheese	Sodium alginate + essential oil (*O. basilicum*, L.; *R. officinalis*, L.; *A. herba alba*; *Asso. M. pulegium*, L.)	Immersion	The growth of *Staphylococci Salmonella* for all types of cheese were completely inhibited	[103]

**Table 3 molecules-27-05604-t003:** Application of smart packaging in food products.

Food Products	Biopolymeric Matrix	Coating Techniques	Results	References
Fish	Chitosan + red cabbage (RC) extract + clove bud oil (CBO)	Edible film	Color turns from purple to deep blue during the growth of fish-spoiling bacteria	[132]
Chicken breast	Roselle anthocyanin + starch	Edible film	Sensitive toward pH changes Changes color from red to yellowish-green when exposed to alkaline environment	[133]
Chicken and fish	Curcumin extract + modified rice starch		Change color from yellow to a reddish-brown or wine-red The LOD of the film was 38.63 μM, LOQ of the film 128.75 μM, and the linear working range was from 0 to 100 μM.	[134]
Grass carp fillets	ĸ-carrageenan + gelatin		Film color changes from yellow to red when exposed to spoil food products	[135]
Chicken breast	Sugarcane bagasse nanocellulose + poly(ether-block-amide) (PEBA) film	Multilayer films	A layer of eight polymer-immobilized pH dyes that changed color, and an outer poly(ethylene terephthalate) film	[136]
Shrimp	Mucilage of *Lallemantia iberica* seed gum (LISG) + curcumin		Strong positive correlation between TVBN content of shrimp	[130]
Fish	Carboxymethyl cellulose (CMC) + cellulose nanofibers (CNF) + shikonin extracted from *Lithospermum erythrorhizon* roots	Multilayer films	The film shows high hydrophobic and antioxidant properties The indicator film showed a reddish-pink for fresh fish (pH = 5.7) and turned blue-violet after 36 h (pH = 6.9)	[137]
-	Anthocyanin + chitosan + cellulose matrix		The CH-Sys changed irreversibly the color from light violet to light yellow in response to different temperature exposition (40 °C until 70 °C), independently of luminosity (0 or 1000 lx)	[138]

**Table 4 molecules-27-05604-t004:** Application of active packaging in food products.

Food Products	Biopolymeric Matrix	Coating Techniques	Results	References
Cheese	Cellulose-based adhesive gelatin + gelatin electrospun fibers + allyl isothiocyanate (AIC)	Adhesive composite	Showed significant antimicrobial activities against *Staphylococcus aureus* ATCC 25923 and Escherichia coli O157:H7 Extended shelf life of cheese from 4 weeks to 8 weeks	[207]
Green chilli	Chitosan + citric acid + glycerol	Film developed as pouches	Maintained the moisture content Reduced wrinkles on the green chillies Preserved the green color of the chili	[160]
Minced beef	Chitosan + kombucha tea	Biocomposite film	Extended shelf life of the minced beef up to 3 days Retarded lipid oxidation Retarded microbial growth Maintained quality of minced beef	[208]
Chicken breast meat	Polyester + chitosan + alginate + black cumin oil (BCO)	Multilayer films	Inhibited *Staphylococcus aureus* and *Escherichia coli* Retained the pH values of chicken breast meat Reduced total aerobic mesophilic in chicken breast meat Lowered psychrotrophic bacteria counts in chicken breast meat	[209]
Lamb meat	Cellulose nanofiber + whey protein + titanium dioxide + rosemary essential oil	Packaging film	Reduced microbial growth Retarded lipid oxidation Reduced lipolysis during storage Extended shelf life from around 6 days to 15 days	[210]
Sliced cooked ham	Green tea extract + oregano essential oil	Packaging film	Reduced microbial growth below the limits of 10^6^ UFC/g Showed good antimicrobial activity against total viable counts (TVC) and lactic acid bacteria (LAB) Lowered *Brochothrix thermosphacta* counts Reduced discoloration of the sliced cooked ham	[198]
Green grape	Agar + zinc oxide nanoparticle synthesized from *Mimusops elengi* fruit extract	Bionanocomposite film	Improved fresh appearance of the green grapes Extended shelf life of green grapes up to 14 to 21 days	[211]
Green bell pepper	Chitosan biguanidine hydrochloride + carboxymethyl cellulose (CMC) + titanium oxide nanoparticles	Nanocomposite film	Resisted weight loss of green bell pepper Reduced spoilage during storage	[212]
Chicken fillet	Gelatin-based nanocomposite + cellulose nanofiber (CNF) + zinc oxide nanoparticles	Nanocomposite film	Reduced water vapor permeability Reduced moisture absorption Provided antibacterial attributes Inhibited growth of Gram-positive strain bacteria	[213]
Strawberry, loquats	Calcium alginate + silver nanoparticles	Edible coating	Maintained acidity and pH over the storage period Prevented loss of soluble solid contents Decreased weight loss Resisted mold growth on the surface of the fruits	[214]

## Data Availability

The data presented in this study are available on request from the corresponding author.

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
