# Peer review of "The Emergence of Edible and Food-Application Coatings for Food Packaging: A Review"

_molecules, 2022, doi:10.3390/molecules27175604_

Round 1
Reviewer 1 Report
The linguistic revision of whole manuscript should be carried out.
The paper should be formatted following Journal guidelines including references format.
Abstract should be reorganized including introduction, aim, methodology, main findings and future remarks.
In Introduction the use of innovative technologies for food packaging should be better introduced.
The aim and the novelty character of the review respect to the other ones present in literature should be better highlighted.
A methodology section describing the bibliographic research criteria should be inserted including a graphical scheme.
The paragraph 2 should be organizd in more paragraphs: a lot informations are reported in confused manner. It should be better organized and clarifed. More details should be given in Figures 1 and 2. Tables reporting the main studied on different packaging described on the basis of literature search should be reported.
Food Safety Issues in Food Packaging should be better described.
Limits, advantages and practical applications should be marked in Conclusion.
Author Response
Journal : Molecules,
Special issues : Using Natural Products as Edible Films and Coatings for Food Applications
Manuscript ID : molecules-1829924
Title : The Emergence of Edible and Food-Application Coatings for Food Packaging: A Review
The authors would like to inform you that the revised manuscript referred to above has corrected the suggestions and constructive comments of the reviewers and editors. The comments and suggestions highlighted by the reviewers were considered in the revised manuscript. The modifications, additions, and corrections appear in the highlights within the article. The following point-to-point is given as follows:
Reviewer 1
1. The linguistic revision of the whole manuscript should be carried out.
- We appreciate your thoughtful, constructive, and insightful comments on this manuscript. The manuscript has been revised accordingly.
2. The paper should be formatted following Journal guidelines including references format.
- The authors have formatted the manuscript and references followed Journal guidelines.
3. Abstract should be reorganized including introduction, aim, methodology, main findings and future remarks.
- We appreciate your constructive comments. The authors have amended the abstract accordingly.
4. In Introduction the use of innovative technologies for food packaging should be better introduced.
- Thank you for the recommendation. The authors have included the corresponding sentences in the introduction section. The use of innovative technologies for food packaging have been explained briefly in Paragraph 2.
5. The aim and the novelty character of the review respect to the other ones present in literature should be better highlighted.
- The authors have highlighted the aim and novelty of the manuscript in the Paragraph 4 of the introduction section.
6. A methodology section describing the bibliographic research criteria should be inserted including a graphical scheme.
- Thank you for the recommendation. The authors inserted a few graphical schemes for each type of food packaging mentioned in the manuscript.
7. Paragraph 2 should be organized in more paragraphs: a lot of information is reported in a confused manner. It should be better organized and clarified. More details should be given in Figures 1 and 2. Tables reporting the main studies on different packaging described on the basis of literature search should be reported.
- Thank you for the insightful comments. The manuscript has been revised accordingly. Besides, the addition of graphical schemes may help to improve the understanding of the readers. The authors have inserted tables to each type of food packaging.
8. Food Safety Issues in Food Packaging should be better described.
- Thank you for the comment and suggestion. The authors have added new information in the Food Safety Issues in Food Packaging
9. Limits, advantages and practical applications should be marked in Conclusion.
- We appreciate the suggestion suggested. The authors have made new amendments in the conclusion section adjusted to fulfil the suggestion given.
Reviewer 2 Report
The manuscript provides a concise overview of the fundamentals of the coating process, including conventional, smart packaging and active packaging. According to this review, researchers will know clearly the specific and detailed theoretical research on food packaging, especially edible film and coating, current problems and future development trends, which is of great significance to improve the efficiency and effect of food packaging. In general, I recommend the acceptance of the manuscript after completely addressing the following concerns.
(1) Page 6, in first line, “Figure 1. Classification of Biodegradable Packaging”. Figure and caption should be in the same page.
(2) Page 11, in 2.3.1. Polysaccharide-based Edible Films, “have the potential to block oxygen…without fostering anaerobic conditions”. Please double check the expression and words.
(3) Page 17, in first line, I recommend that the words in the diagram start with a capital letter rather than all capital letters, as in Figures 1 and 2.
(4) Page 25, in 3. Food Safety Issues in Food Packaging part. I'm interested in smart packaging and active packaging. But I didn't find any discussion about their safety issues. More discussions are needed.
Author Response
Reviewer 2
The manuscript provides a concise overview of the fundamentals of the coating process, including conventional, smart packaging and active packaging. According to this review, researchers will know clearly the specific and detailed theoretical research on food packaging, especially edible film and coating, current problems and future development trends, which is of great significance to improve the efficiency and effect of food packaging. In general, I recommend the acceptance of the manuscript after completely addressing the following concerns.
1. Page 6, in the first line, “Figure 1. Classification of Biodegradable Packaging”. Figure and caption should be on the same page.
- Thank you for the comment. The figure and caption has been included on the same page.
2. Page 11, in 2.3.1. Polysaccharide-based Edible Films, “have the potential to block oxygen…without fostering anaerobic conditions”. Please double check the expression and words.
- The authors have checked and updated the words and expressions accordingly.
3. Page 17, in first line, I recommend that the words in the diagram start with a capital letter rather than all capital letters, as in Figures 1 and 2.
- Thank you for the suggestion. The author has revised Figure 3.
4. Page 25, in 3. Food Safety Issues in Food Packaging part. I'm interested in smart packaging and active packaging. But I didn't find any discussion about their safety issues. More discussions are needed.
- Thank you for the comments. The authors have included the discussion regarding the safety issues of smart and active packaging in the Food Safety Issues in Food Packaging section.
Reviewer 3 Report
This review article is an exceptional contribution to the field of materials for packaging. This review is comprehensive and well written. I only have few minor suggestions for further improvement:
1. Table 1 is very long; I suggest to split it in three smaller tables dedicated to biodegradable, smart and active packaging to be moved to the relevant sections.
2. The section on safety is very dry and needs to be expanded.
3. Quite surprisingly, this review paper lacks of illustrations. Some schemes are reported but they are not enough. High-quality figures are necessary in a good review paper, please improve this aspect.
Author Response
Reviewer 3
This review article is an exceptional contribution to the field of materials for packaging. This review is comprehensive and well written. I only have few minor suggestions for further improvement:
1. Table 1 is very long; I suggest to split it in three smaller tables dedicated to biodegradable, smart and active packaging to be moved to the relevant sections.
- The authors appreciate the suggestion given. Table 1 has been split into its relevant sections.
2. The section on safety is very dry and needs to be expanded.
- Thank you for the comment and suggestion. The authors have added new information in the Food Safety Issues in Food Packaging
3. Quite surprisingly, this review paper lacks illustrations. Some schemes are reported but they are not enough. High-quality figures are necessary in a good review paper, please improve this aspect.
- Thank you for the comment and suggestion. The illustrations in the manuscript have been improvised accordingly.
Round 2
Reviewer 1 Report
The paper is suitable for publication